# Impact of Catheter Ablation on Brain Microstructure and Blood Flow Alterations for Cognitive Improvements in Patients with Atrial Fibrillation: A Pilot Longitudinal Study

**DOI:** 10.3390/jcm11154346

**Published:** 2022-07-26

**Authors:** Yasuko Tatewaki, Tatsushi Mutoh, Hirokazu Sato, Akiko Kobayashi, Tomoko Totsune, Benjamin Thyreau, Atsushi Sekiguchi, Taizen Nakase, Tetsuo Yagi, Yasuyuki Taki

**Affiliations:** 1Department of Aging Research and Geriatric Medicine, Institute of Development, Aging and Cancer, Tohoku University, Sendai 980-8575, Japan; tomoko.totsune.e5@tohoku.ac.jp (T.T.); benjamin.thyreau.a5@tohoku.ac.jp (B.T.); taizen.nakase.a4@tohoku.ac.jp (T.N.); yasuyuki.taki.c7@tohoku.ac.jp (Y.T.); 2Department of Geriatric Medicine and Neuroimaging, Tohoku University Hospital, Sendai 980-8575, Japan; 3Department of Surgical Neurology, Research Institute for Brain and Blood Vessels-AKITA, Akita 010-0874, Japan; 4Division of Cardiology, Sendai City Hospital, Sendai 982-8502, Japan; hirokazu@med.tohoku.ac.jp (H.S.); tetsuo.yagi@nifty.com (T.Y.); 5Sendai Seiryo Clinic, Sendai 980-0801, Japan; 6Department of Functional Brain Imaging, Institute of Development, Aging and Cancer, Tohoku University, Sendai 980-8575, Japan; akiko.kobayashi.q6@alumni.tohoku.ac.jp; 7Smart-Aging International Research Center, Tohoku University, Sendai 980-8575, Japan; 8Department of Behavioral Medicine, National Institute of Mental Health, National Center of Neurology and Psychiatry, Tokyo 187-8551, Japan; asekiguchi@ncnp.go.jp

**Keywords:** atrial fibrillation, catheter ablation, cerebral blood flow, cognitive function, magnetic resonance imaging, surface-based morphometry

## Abstract

Atrial fibrillation (AF) predisposes patients to develop cognitive decline and dementia. Clinical and epidemiological data propose that catheter ablation may provide further benefit to improve neurocognitive function in patients with AF, but the underlying mechanism is poorly available. Here, we conducted a pilot prospective study to investigate whether AF ablation can alter regional cerebral blood flow (rCBF) and brain microstructures, using multimodal magnetic resonance imaging (MRI) technique. Eight patients (63 ± 7 years) with persistent AF underwent arterial-spin labeling (ASL) perfusion, 3D T1-structural images and cognitive test batteries before and 6 months after intervention. ASL and structural MR images were spatially normalized, and the rCBF and cortical thickness of different brain areas were compared between pre- and 6-month post-treatment. Cognitive–psychological function was improved, and rCBF was significantly increased in the left posterior cingulate cortex (PCC) (*p* = 0.013), whereas decreased cortical thickness was found in the left posterior insular cortex (*p* = 0.023). Given that the PCC is a strategic site in the limbic system, while the insular cortex is known to play an important part in the central autonomic nervous system, our findings extend the hypothesis that autonomic system alterations are an important mechanism explaining the positive effect of AF ablation on cognitive function.

## 1. Introduction

Atrial fibrillation (AF) is the most common cardiac arrhythmia and has a disease prevalence that increases with age [1]. Several epidemiological studies reported that AF is a risk factor for cognitive decline and dementia in elderly populations [2,3]. AF is strongly associated with an increased risk of stroke and can cause vascular dementia.

However, independent of cerebral infarction, AF is also reported to increase the risk of dementias including Alzheimer’s disease, suggesting that AF itself has additional negative effects on cognitive function [4]. Several potential mechanisms other than stroke have been proposed to explain the association between AF and cognitive decline, including chronic brain hypoperfusion due to reduced stroke volume, microembolism, cerebral microbleeds, vascular inflammation, and brain atrophy [5,6].

Catheter ablation therapy is an option for the surgical cure of AF, and patients with AF can achieve complete rhythmic control through this procedure. Although emerging research suggests that catheter ablation might mitigate cognitive decline, reduce the risk of AD, and improve cognitive function [4,7,8], the exact neural basis through which ablation therapy may exert such effects has not been well investigated.

Magnetic resonance imaging (MRI) is a non-invasive imaging modality that enables brain structure to be delineated, while arterial spin labeling (ASL) is an MRI perfusion method that can provide information on regional brain perfusion [9]. The quantitative STAR labeling of arterial regions (QUASAR)-ASL method using dynamic acquisitions and a model-based quantitation approach can improve the reliability of the quantitative measurement of regional cerebral blood flow (rCBF) and whole CBF [9,10]. Additionally, advanced image acquisition techniques can be combined with computer-aided image analysis to visualize regional structural or functional features of the brain, allowing for the assessment of interactions with socio-physiological, pathological, and drug-induced factors.

However, it remains unclear how AF directly affects brain cognitive functions, and the improvement benefit from the ablation therapy in older adults with healthy aging has not been reported. Therefore, we conducted a pilot prospective study to investigate whether AF ablation can alter rCBF and brain microstructures, using multimodal MRI technique. Such geometric information in brain alterations might clarify the pathogenesis linking AF with cognitive decline.

## 2. Materials and Methods

### 2.1. Participants

This study enrolled eight right-handed patients with persistent AF (average age of 63.4 ± 7.3 years; range: 49–73) who visited the division of Cardiology, Sendai City Hospital between 1 February 2016 and 31 July 2019. Eligible patients were those with persistent AF for whom a clinical decision was made to perform ablation therapy. The exclusion criteria for all patients were: (1) neuro-psychological disorders, (2) coexisting severe medical conditions or terminal diseases (e.g., stroke, Parkinson’s disease, thyroid/parathyroid disease, and cancer) that may influence the results of the imaging and cognitive studies, (3) difficulty in obtaining written informed consent, (4) difficulty in undergoing brain MRI (pacemaker-implantation, coronary artery stenting, other metal device implantation, or claustrophobia), (5) difficulty in undergoing cognitive tests (amblyopia, deafness, orthopedic impairment) and (6) failure to acquire rhythm control through ablation therapy.

All experiments were performed in accordance with the Declaration of Helsinki, and the study protocol was approved by the Ethics Committee of Tohoku University Graduate School of Medicine (2018-1-760) on 1 January 2019. This trial was registered in the University hospital Medical Information Network Clinical Trials Registry (UMIN-CTR) on 10 July 2016 (UMIN000023007). Written informed consent was obtained from all participants.

### 2.2. Data Collection and Protocol

For this single-arm pilot study, all data and information regarding the patients’ clinical courses were collected and reviewed from medical records maintained by Sendai City Hospital. Before and 6 months after the latest ablation therapy by which rhythm control was acquired, cognitive–psychological tests, cardiac function with echocardiography, serum brain natriuretic peptide (BNP) concentrations, regional cerebral blood flow with QUASAR-ASL MRI, and regional cortical thickness with structural MRI were measured. The control group included seven subjects of similar age (63.0 ± 6.9 years; range: 52–74) who had no arrhythmia and no organic or psychological disorders. They underwent cognitive–psychological tests and MRI images for comparisons with the AF group.

### 2.3. Interventional Procedures for AF Ablation

All procedures were performed via peripheral venous access and transseptal access across the interatrial septum to access the left atrium (LA). All the patients with AF underwent catheter ablation using radiofrequency (RF) energy or cryoballoon energy. A multipolar diagnostic catheter was placed in the coronary sinus. RF therapy procedures were circumferential pulmonary vein isolation (PVI)-guided by a three-dimensional mapping system (CARTO3™; Biosense Webster, Irvine, CA, USA) using an irrigated-tip contact force-sensing RF ablation catheter (SmartTouch Surround Flow™; Biosense Webster). A circular mapping catheter was used to guide PVI and LA ablations. Additional regions (e.g., roof line, bottom line, cavotricuspid isthmus line) were left to the discretion of the individual operator. The procedure was deemed complete when bidirectional entrance (the stable absence of conduction into the PV from the LA) and exit block (the stable absence of conduction from the PV into the LA, either spontaneous or during pacing from the circular mapping catheter positioned at the PV ostium) was achieved in all PVs. The decision on whether to perform mapping and ablation in sinus rhythm after electrical cardioversion or in AF was left to the electrophysiologist performing the procedure.

### 2.4. Cognitive–Psychological Tests

Given the possibility that AF causes dementia and affects memory or executive function, several cognitive tests were administered before and 6 months after the therapy. These included the Japanese version of the Mini Mental State Examination (MMSE-J), Digit Span and Symbol Tests (Wechsler Adult Intelligence Scale—4th Edition; WAIS-IV), Trail Making Test (TMT-A and B), and Rey Auditory Verbal Learning Test (RAVLT). The MMSE was used for screening for dementia, the Digit Span Test to measure working memory, and the Digit Symbol Test to assess processing speed. TMT-A and B were used to assess visual searching, attention, and mental flexibility. The RAVLT was used to assess verbal memory and included measurements of the total numbers of words immediately recalled over the first five trials (T1 − T5) and delayed recall (T6) of the original list. Based on a previous study, we calculated the recognition, total recall (T1 + T2 + T3 + T4 + T5), learning (T5 − T1), and forgetting (T5 − T6) scores from the results of the RAVLT for each participant. These tests were administered by a trained examiner. The results of the tests were not disclosed to the participants until the data collection was finished. In addition, considering that depressive and anxiety states can affect cognitive performance, we also administered self-reported questionnaires: the Center for Epidemiologic Studies Depression Scale (CES-D) and the State-Trait Anxiety Inventory (STAI). STAI-Y1 and STAI-Y2 can be used to measure state anxiety and trait anxiety, respectively.

### 2.5. Echocardiography

Before and 6 months after ablation therapy, echocardiography studies were performed by experienced sonographers blinded to the results of the cognitive tests and the MRI. Measurements were made from an average of five consecutive cycles of AF. The ejection fraction was determined in B-mode using Simpson’s method. Left atrial and left ventricular maximum diameter/volume and cardiac output were evaluated for each patient with AF.

### 2.6. Brain Image Acquisition

A 3.0-T MRI scanner (Intera Achieva 3.0T Quasar Dual, Philips, Amsterdam, The Netherlands) and an 8-element head coil were used to acquire a three-dimensional high resolution T1-weighted magnetization-prepared rapid acquisition gradient echo structural image with the following parameters: matrix = 240 × 240, repetition time (TR) = 8.70 ms, echo time (TE) = 3.1 ms, flip angle = 8°, field of view (FOV) = 256 × 256 × 180 mm, 162 slices, voxel size = 0.7 × 0.7 × 0.7 mm, and scan duration 5 min 15 s. Pulsed ASL brain perfusion images were collected using a QUASAR implementation [9,10], with the participants instructed to keep their eyes closed. The position of the labeling slice was determined as the fourth of the seven slices positioned on the body of the corpus callosum in the coronal scout view. The parameters used were: matrix = 64 × 64, TR = 300 ms, TE = 22 ms, FOV = 240 × 240 mm, 7 slices, slice thickness = 7.0 mm, interslice gap = 2.0 mm, SENSE factor = 2.5, signal averages = 84, and scan duration 5 min 52 s. An R1 map was obtained as part of the ASL scan. The technical details of this scan were described in a previous study [9]. The following constants were used in the CBF calculation: T1 of arterial blood, 1.65 s; inversion efficiency, 95%; blood–brain partition coefficients for GM and WM, 0.98 and 0.82, respectively.

Preprocessing of structural and ASL data for voxel-based analysis or surface-based morphometry was performed using Statistical parametric mapping 12 (SPM12; The Wellcome Centre for Human Neuroimaging; https://www.fil.ion.ucl.ac.uk/spm/ (accessed on 1 July 2022)) implemented in Matlab. Spatially normalized regional CBF maps with partial volume correction of regional gray matter volume (rGMV) and measurements of cortical thickness throughout the whole brain were obtained for statistical analysis. Details of the procedures are presented elsewhere [11] and are described in the Appendix A.

### 2.7. Statistical Analysis

For the demographic data, continuous variables are presented as mean ± standard deviation (SD). Comparisons between the AF and normal control groups were conducted using *t* tests or Mann–Whitney *U* tests. Changes in clinical data and cognitive tests after ablation therapy were evaluated using paired *t* tests or Wilcoxon signed-rank tests to compare baseline conditions with those 6 months after ablation. Statistical analysis was performed using JMP Pro 15 (SAS Institute Inc., Cary, NC, USA) with a significance threshold of *p* < 0.05. The false-discovery rate (FDR) correction was used to correct for multiple comparisons.

Brain areas showing rCBF changes after ablation therapy were identified using voxel-wise comparisons of ASL based rCBF maps before and 6 months after ablation therapy, with differences evaluated with paired *t* tests. The significance threshold was set as a cluster-level FDR-corrected *p* value lower than 0.05, with this correcting for multiple comparisons across the whole brain according to random field theory.

For surface-based morphometry (SBM) to explore which brain areas showed cortical thickness changes after ablation therapy, cortical morphometric maps of the left and right hemispheres were separately analyzed using paired *t* tests, with age and gender as covariates using the CAT12 (Computational Anatomy Toolbox; https://neuro-jena.github.io/cat// accessed on 1 July 2022)). Results were considered significant only if the peak-level FDR-corrected *p* value was lower than 0.05.

## 3. Results

### 3.1. Patient Characteristics

The participants’ demographic and clinical data are presented in Table 1. Of the eight patients with AF, regarding the time from diagnosis to AF ablation, six were less than one-year and two were one to three years in duration. The baseline CHADS_2_ score was one patient at 0 and seven patients at 1. At baseline, five patients were prescribed anticoagulants including warfarin and direct oral anticoagulant (DOAC), one was on an antiplatelet drug, and two were on anti-arrhythmic medications. No patients were prescribed antidepressants. We interviewed seven patients who were available for post-intervention long-term follow-up. Antiarrhythmic drugs were prescribed for three of them, including two patients who relapsed more than six months after AF ablation. The five patients were continuously on DOAC after ablation therapy. According to the results of the standard neurological examination, none of the participants showed any focal neurological symptom. Visual review of conventional T2-weighted FLAIR MRI found no cerebral infarction/chronic ischemic change or hemorrhage in patients before and 6 months after AF ablation therapy. Echocardiography 6 months after ablation therapy revealed no significant change in cardiac output or other cardiac parameters compared with baseline values. However, after intervention the serum concentration of BNP was significantly lower than at baseline (*t*(7) = −2.228, r = 0.64; *p* = 0.034).

### 3.2. Longitudinal Changes in Cognitive Function after a 6-Month AF Ablation Therapy

Comparison of baseline values between the AF group and normal controls revealed significant differences in the scores of MMSE (*t*(13) = 2.918, r = 0.63; *p* = 0.02), STAI-Y1 (*t*(13) = −3.783, r = 0.72; *p* = 0.003) and STAI-Y2 (*t*(13) = −3.155, r = 0.66; *p* = 0.008) (Table 2). The AF group showed lower general cognitive function and higher state and trait anxiety than the normal controls.

Six months after ablation therapy, the AF group showed significantly higher scores than at baseline in the MMSE (*t*(6) = 2.50, r = 0.71; *p =* 0.023), WAIS digit symbol (*t*(6) = 3.29, r = 0.80; *p* = 0.008), and total recall of RAVLT (*t*(6) = 2.50, r = 0.71; *p* = 0.023), and significantly lower scores in the STAI-Y1 (*t*(7) = −3.65, r = 0.81; *p* = 0.004). AF ablation therapy affected the psychological state of anxiety and cognitive functions including executive functions and verbal memory. The scores in the other psychological tests did not show significant changes after ablation therapy.

### 3.3. Changes in Brain Perfusion after a 6-Month AF Ablation Therapy

We calculated the whole cerebral gray matter CBF ratio normalized by the cerebral white matter CBF value from the QUASAR-MRI maps. There was no significant difference in the baseline whole cerebral gray matter CBF ratio between the AF group and normal controls (1.59 ± 0.07 in the AF group versus 1.64 ± 0.07 in the normal control group; *p* = 0.61). In the AF group, there was no significant difference in the whole cerebral gray matter CBF ratio between baseline and 6 months post-AF ablation (*p* = 0.77).

In the AF group, voxel-based paired *t* tests conducted using SPM12 showed significantly increased rCBF in the left posterior cingulate gyrus (PCC) after ablation therapy, particular in the retrosplenial cortex (MNI coordinates at peak voxel: *X* = −9 mm, *Y* = −45.5 mm, *Z* = 0 mm; *t* = 10.8, cluster size = 205; FDR corrected *p* = 0.013) (Figure 1).

### 3.4. Microanatomical Changes in Cortical Thickness after a 6-Month AF Ablation Therapy

We used the aforementioned SBM approach to evaluate longitudinal changes in regional cortical thickness associated with ablation therapy. The SBM method is often more sensitive for intra-subject serial change in brain structure than the voxel-based morphometry (VBM) algorithm implemented in SPM12. In the AF group, we found a significant decrease in cortical thickness in the left posterior insular cortex 6 months after ablation therapy (MNI coordinates at peak voxel: *X* = −38 mm, *Y* = −3 mm, *Z* = −8 mm; *t* = 13.36, cluster size = 60; FDR corrected, *p* = 0.023) (Figure 2). SBM did not detect any region showing an increase in cortical thickness after ablation therapy.

## 4. Discussion

In this study, we investigated whether changes in cognitive improvement occurring at 6 months following AF ablation were associated with the MRI-measured parameters of rCBF and cortical thickness in the brain. The results showed increased rCBF in the left PCC (retrosplenial region) and decreased cortical thickness in the left posterior insular cortex after the intervention. To the best of our knowledge, this is the first report to use multimodal MRI sequences to clarify the neural basis of the cognitive improvement effect of catheter AF ablation.

It is broadly known that a relationship between AF and cognitive impairment exists. Patients with AF may present with lower cognitive functioning than non-AF subjects, and AF can increase the risk of dementia, including AD [2,3]. In the present study, patients with AF tended to have lower MMSE and higher anxiety state scores than healthy subjects at baseline, findings that are consistent with previous reports. In addition, our results showed improvements in a broad spectrum of cognitive domains represented by the MMSE, WAIS digit symbol, RAVLT verbal memory, and STAI scores after ablation treatment. Several interventional studies have mentioned the effects of catheter AF ablation on neuropsychological functions. Jin and colleagues [4] reported a significant improvement in cognitive function (assessed using the Montreal Cognitive Assessment) in an AF ablation group at 3 months and 1 year after the intervention, with the cognitive improvement being more pronounced among those patients with a mild cognitive impairment at baseline. More recently, a prospective study enrolled 74 Japanese patients who underwent AF ablation and reported that scores for the MMSE, immediate recall, delayed recall, constructional visuospatial ability, and TMT were significantly improved 6 months after ablation therapy [7]. However, contrary to such reports suggesting long-term protective effects on cognitive function, Kochhauser et al. [12] found no significant cognitive change associated with AF ablation. Our results are consistent with the positive reports, and we found that cognitive decline associated with AF pathophysiology seemed to be recoverable.

Previous reports indicating that AF ablation may facilitate cognitive recovery from AF-related pathogenesis by restoring sinus rhythm are subject to limitations, and few mechanistic studies have yet been attempted. In the present study, the left retrosplenial cortex, which is a part of the PCC, was identified as a location showing blood flow changes associated with AF ablation. The retrosplenial cortex is an important part of the limbic system and is deeply involved in cognitive functions including episodic memory, navigation, imagination, and planning for the future, and is known to be a region where functional decline occurs from the early stages of AD [13]. Therefore, it seems reasonable to assume that improvement in rCBF in the retrosplenial cortex could contribute to improved cognitive function, and we believe that this result reveals part of the mechanism explaining the linkage between AF and cognitive impairment. Previously, several studies have suggested chronic brain hypoperfusion as a plausible mechanism linking AF to cognitive decline [6]. AF causes cerebral hypoperfusion through beat-to-beat variability and an overall reduction in cardiac output owing to the lack of atrioventricular synchrony [14]. Although few studies evaluating serial changes in CBF after AF ablation have been conducted, there is one report of improved cognitive function after pacemaker implantation (another treatment for AF). Using single photon emission computed tomography with 99mTc-hexamethylpropylene amine oxime, this study found increased rCBF in the right inferior frontal, left superior frontal, and left temporal cortex, with the increased rCBF corresponding to improvement in cardiac output [15]. Although in our patients, significant restoration of cardiac output was not clear because the baseline of their cardiac output was not impaired, some cognitive domains and rCBF in the left retrosplenial cortex were improved after AF ablation, independent of cardiac output. Therefore, we speculate that AF-related adverse factors as well as cardiac output may be involved in the recovery of cognitive function after ablation therapy.

In the current study, we used cortical surface-based morphometry in the CAT12 toolbox to quantify cortical thickness in patients with AF, to detect precise changes in regional cortical structure after ablation therapy. Regional changes in brain structure, including cortical thickness, generally imply functional differences corresponding to localization in the brain. For example, it is known that the regional gray matter volume of the visual cortex is reduced in patients with glaucoma, showing visual field defects [16]. We found decreased cortical thickness in the left posterior insular cortex, which is well known as an autonomic center of the central nervous system [17,18]. Therefore, our results could indicate that AF ablation may affect the functioning of the central autonomic system. In patients with AF, overload of the autonomic neurons causes excessive release of neurotransmitters, which can lead to triggered evoking and re-entry of electrical potentials at multiple heart sites to initiate and maintain AF [19]. Modulation of the cardiac autonomic system, denervation of the extrinsic cardiac sympathetic nervous system, and ganglionated plexi ablation are broadly employed for efficient radical therapy for AF [20]. Our results indicate that the restoration of sinus rhythm by ablation therapy may affect the autonomic centers of the central system in the posterior insular cortex through direct injury to the ganglionated plexi by the AF ablation procedure, or through correction of the overload of both the extrinsic and intrinsic autonomic nervous system attributed to rhythm control. Thus, alteration of the cardiac autonomic system caused by the AF ablation procedure could contribute to recovery of rCBF in the PCC, followed by a protective effect on cognitive function.

It is also known that the autonomic nervous system plays a role in the vascular reactivity of cerebral blood vessels. Sympathetic activity can decrease CBF or attenuate a CBF increase [21]. Sympathetic nerve activity can limit cerebral vasodilatation during severe hypertension, hypoxia, and hypercapnia. A high cognitive load requires excessive CBF compared with resting conditions, which is provided through a process called “neuro-vascular coupling”. We suppose that AF patients might be prevented from sufficiently increasing CBF in response to cognitive workload because of hyperactivity of the autonomic nervous system. We hypothesize that the restoration of sinus rhythm by ablation improves vascular reactivity by correcting autonomic hyperactivity, which in turn improves blood flow in the PCC, because the PCC is a border zone between the posterior and anterior circulation, and is the part most vulnerable to dysfunction of the autoregulation system. This improved vascular reactivity may then have contributed to the improvement in cognitive function. Although ablation-induced changes in factors such as norepinephrine in serum will need to be investigated to provide evidence for this hypothesis, the present study could imply that rapid changes in the autonomic nervous system may be an important candidate mechanism for explaining AF-related cognitive decline.

Recently, the effectiveness of catheter ablation as a first-line treatment for AF has received much attention, and it is reported that first-line ablation therapy significantly reduced the recurrence of AF as compared to first-line antiarrhythmic drugs [22]. The present results may indicate the direct impact on brain function as an additional benefit of ablation therapy and may encourage the indication of ablation therapy as a first-line treatment for patients with AF. The new strategy of “early AF ablation” could contribute to maintaining cognitive function and quality of life in a super-aging society. 

### Limitations

This pilot study has several limitations with regard to the small sample size and lack of a non-interventional control within the AF patients. Therefore, our results may be subject to selection bias and may not only necessarily demonstrate the effect of AF ablation therapy. In addition, the patients enrolled in this study have normal cardiac function at baseline. This study could not examine the possibility that an increase in cardiac output could be an important mechanism of Ablation-depending cognitive recovery in AF patients that suffer from concomitant heart failure with impaired cardiac output. We could not adjust several factors related to lifestyles (e.g., socio-economic level, psychological effects, and physical activity), which may also affect cognitive functions. Nevertheless, present longitudinal, prospective data derived from the multimodality MRI technique compared with age-matched normal database with at least impart provided the most plausible brain areas to improve autonomic–cognitive functions after the ablation therapy.

## 5. Conclusions

Our study showed that improved rCBF in the left PCC and decreased cortical thickness in the left posterior insular cortex corresponded to restoration of cognitive function after catheter ablation therapy in patients with persistent AF, independent of improvement in hemodynamic parameters (e.g., cardiac output). The PCC is a strategic site in the limbic system, while the insular cortex is known to play an important part in the central autonomic nervous system. Our results extend the hypothesis that autonomic system alterations are an important mechanism explaining the positive effect of AF ablation on cognitive function. Further experiments to clarify the neural network regulating the automomic/cognitive functions will strengthen the present findings.

## Figures and Tables

**Figure 1 jcm-11-04346-f001:**
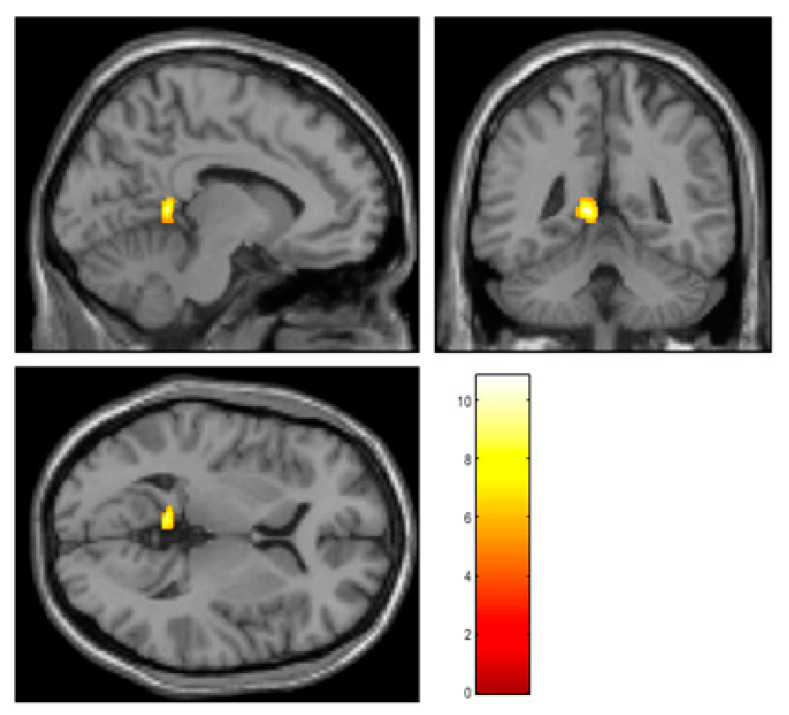
The results of voxel-based paired *t* tests using SPM12. The rCBF was significantly increased in the left retrosplenial cortex 6 months after AF ablation. The retrosplenial cortex is known as a critical part of the limbic system and is deeply associated with cognition. The color bar indicates *t* value.

**Figure 2 jcm-11-04346-f002:**
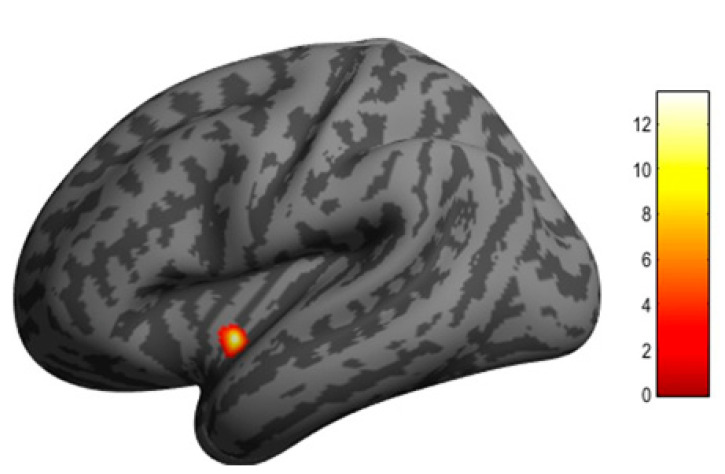
Clusters in the left posterior insular cortex where cortical thickness was significantly decreased 6 months after AF ablation according to surface-based morphometry with CAT12. The insular cortex is a central part of the autonomic neural system. The color bar indicates *t* value.

**Table 1 jcm-11-04346-t001:** The patients’ clinical characteristics and indices of cardiac function before and after AF ablation therapy.

	Baseline	After AF Ablation	*p* Value
**Gender (M/F)**	2/6	-	
**Age (years)**	63.4 ± 7.3 (49–72)	-	
**Heart rate (bpm)**	82.6 ± 29.3 (53–136)	72.3 ± 15.5 (59–88)	0.264
**LAD (mm)**	41.4 ± 6.6 (33.1–53.6)	40.3 ± 6.7 (42.9–53.0)	0.183
**LVDd (mm)**	46.7 ± 4.4 (40.7–52.6)	45.8 ± 4.5 (41.4–47.5)	0.116
**LVDs (mm)**	30.9 ± 4.0 (25.8–36.5)	30.1 ± 3.6 (25.6–36.5)	0.169
**EF (%)**	62.9 ± 6.0 (52.1–68.4)	63.2 ± 4.9 (52.6–68.8)	0.435
**LA volume (mL)**	75.8 ± 16.5 (50.7–99.0)	74.6 ± 25.3 (55.8–124.0)	0.405
**LA volume index (mL/m2)**	45.1 ± 8.0 (32.5–54.3)	43.5 ± 11.3 (29.5–63.9)	0.277
**CO (L/min)**	4.5 ± 0.8 (3.2–5.6)	4.3 ± 0.9 (3.1–5.7)	0.603
**CI (L/min/m2)**	2.7 ± 0.6 (1.7–3.3)	2.6 ± 0.4 (1.9–3.0)	0.720
**Serum BNP level (pg/mL)**	83.2 ± 71.9 (26.1–229.3)	37.1 ± 29.0 (7.5–92.1)	0.034 *

Data are expressed as mean ± standard deviation. *p*, differences in values before and after ablation therapy according to *t* test; AF, atrial fibrillation; BNP, brain natriuretic peptide; CI, cardiac index; CO, cardiac output; LAD, left atrial diameter; LVDs, systolic left ventricular diameter; LVDd, diastolic left ventricular diameter; EF, ejection fraction. Significance, by *t* test: *, *p* < 0.05.

**Table 2 jcm-11-04346-t002:** The results of neuropsychological tests in normal controls and before and after AF ablation therapy.

	Objectives	NC Group	*p* Value (NC vs. Baseline)	AF Group		*p* Value (Pre vs. Post)
				Pre	Post	
**Gender (M/F)**		2/5	0.876	2/6		
**Age**		63.0 ± 6.9	0.921	63.4 ± 7.3		
**MMSE**	Total cognitive function	29.6 ± 0.5	0.015 *	27.6 ± 1.7	28.9 ± 1.5	0.023 *
**WAIS** **digit symbol**	Executive function	12.7 ± 1.0	0.157	11.3 ± 3.3	11.9 ± 4.0	0.008 *
**WAIS** **digit span**	Working Memory	9.6 ± 0.7	0.666	10.0 ± 2.3	10.4 ± 3.5	0.267
**TMT-A (s)**	Executive function	66.6 ± 7.8	0.205	80.9 ± 25.9	73.7 ± 16.4	0.099
**TMT-B (s)**	Executive function	87.2 ± 16.5	0.393	106.5 ± 43.6	103.7 ± 33.0	0.148
**RAVLT**	Verbal memory					
**Recognition**		14.4 ± 1.0	0.032	11.1 ± 3.5	13.4 ± 1.3	0.078
**Total recall**	Immediate memory	48.5 ± 3.0	0.066	39.3 ± 14.1	44.6 ± 15.2	0.023 *
**Learning**		7.0 ± 0.6	1.000	7.0 ± 2.3	4.4 ± 2.0	0.954
**Forgetting**	Delayed recall	2.5 ± 0.6	0.500	1.9 ± 2.3	1.4 ± 2.8	0.427
**CES-D**	Depressive state	6.8 ± 3.6	0.084	15.9 ± 10.8	13.5 ± 6.2	0.287
**STAI**	Anxiety state					
**Y1**		32.6 ± 3.2	0.003 *	48.5 ± 9.4	38.5 ± 8.8	0.004 *
**Y2**		33.0 ± 4.4	0.008 *	51.8 ± 12.8	47.8 ± 10.7	0.128

Data are expressed as mean ± standard deviation. *p*, differences in values between normal controls and baseline of AF according to t test and differences in values before and after ablation therapy according to paired *t* test, AF, atrial fibrillation; RAVLT, Ray Auditory Verbal Learning Test; MMSE, Mini mental state examination; CES-D, The Center for Epidemiologic Studies Depression Scale; TMT, Trail making test; STAI, State-Trait Anxiety Inventory; WAIS, Wechsler Adult Intelligence Scale-Fourth Edition. Significance: *, *p* < 0.05.

## Data Availability

Anonymized data of patients are available from the corresponding author on reasonable request.

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
