# Peer review of "Impact of Catheter Ablation on Brain Microstructure and Blood Flow Alterations for Cognitive Improvements in Patients with Atrial Fibrillation: A Pilot Longitudinal Study"

_jcm, 2022, doi:10.3390/jcm11154346_

Round 1

Reviewer 1 Report

The article is well written. The study has a good design. The paper is logically divided into sections and subsections. The references cited are relevant and adequate. The work has a degree of novelty and the subject needs further study. There are some places where wording may be improved. In my opinion, this paper can be recommended for publication after minor revision.

1. Please add pharmacotherapy to the baseline characteristics, particularly anticoagulation and antiarrhythmic drugs.

2. If available, explore the duration of AF - the time from diagnosis to AF ablation; the results might be biased depending on the duration of AF. 

3. In my opinion, it might be interesting to discuss the new strategy of "early AF ablation" - and its potential benefits.

(e.g. Imberti JF, Ding WY, Kotalczyk A, Zhang J, Boriani G, Lip G, Andrade J, Gupta D. Catheter ablation as first-line treatment for paroxysmal atrial fibrillation: a systematic review and meta-analysis. Heart. 2021 Oct;107(20):1630-1636. doi: 10.1136/heartjnl-2021-319496. Epub 2021 Jul 14. PMID: 34261737.)

Author Response

The article is well written. The study has a good design. The paper is logically divided into sections and subsections. The references cited are relevant and adequate. The work has a degree of novelty and the subject needs further study. There are some places where wording may be improved. In my opinion, this paper can be recommended for publication after minor revision.

Author’s Response: Thank you very much for reviewing our manuscript and for valuable important comments. We have revised our manuscript in accordance with the reviewer’s comments.

  1. Please add pharmacotherapy to the baseline characteristics, particularly anticoagulation and antiarrhythmic drugs.

Author’s Response:

Thank you for the reviewer’s important comment. We inserted the following sentences in the Results section. 

At baseline, 5 patients were prescribed anticoagulants including warfarin and direct oral anticoagulant, and 1 was on antiplatelet drug, and 2 were on anti-arrhythmic medications. No patients were prescribed antidepressants.

  1. If available, explore the duration of AF - the time from diagnosis to AF ablation; the results might be biased depending on the duration of AF. 

Author’s Response:

Thank you for the comment. According to the reviewer’s advice, we added the information about AF duration in the Results section.

Of the eight patients with AF, regarding the time from diagnosis to AF-ablation, 6 were less than one-year and 2 were one to three years in duration.”

  1. In my opinion, it might be interesting to discuss the new strategy of "early AF ablation" - and its potential benefits. (e.g. Imberti JF, Ding WY, Kotalczyk A, Zhang J, Boriani G, Lip G, Andrade J, Gupta D. Catheter ablation as first-line treatment for paroxysmal atrial fibrillation: a systematic review and meta-analysis. Heart. 2021 Oct;107(20):1630-1636. doi: 10.1136/heartjnl-2021-319496. Epub 2021 Jul 14. PMID: 34261737.)

Author’s Response:

We greatly appreciated the reviewer’s important comment. The effectiveness of the Early AF ablation strategy is a strongly interesting issue clinically and epidemiologically. We inserted one paragraph about this issue in the Discussion section and added a new citation [22].

Recently, the effectiveness of catheter ablation as a first-line treatment for AF has been much attention and it is reported that first-line ablation therapy significantly reduced the recurrence of AF as compared to first-line antiarrhythmic drugs [22]. The present results may indicate the direct impact on brain function as an additional benefit of ablation therapy and encourage the indication of ablation therapy as a first-line treatment for patients with AF. The new strategy of "early AF ablation” could contribute to maintaining cognitive function and quality of life in a super-aging society.”

Reviewer 2 Report

This interesting manuscript by Tatewaki et. al investigated the impact of pulmonary vein isolation (PVI) on cognitive-psychological function as well as regional cerebral blood flow (rCBF) and brain microstructures. The authors find an increased rCBF in the left posterior cingulate gyrus, and decreased cortical thickness in the left posterior insular cortex, accompanied by restored cognitive function, 6 months after PVI procedure. Thus, the authors conclude, that autonomic system alterations after catheter ablation may be a potential mechanism explaining the positive effects on cognitive function.

Despite its small size, this is an interesting study, which generates a valid hypothesis.

I have some comments, that should be addressed by the authors in a revised version of the manucript:

1)      Could the authors provide information about the exact duration of the AF disease? Did all patients profit equally from the procedure, regardless of the AF duration?

2)      Could the authors povide information about the medication of the patients, particularly possible anti-arrhythmic drugs, anticoagulation and antidepressives? Did any patients receive additional anti-arrhythmic drugs after the PVI procedure.

3)      Did any patients have AF recurrences after the PVI treatment? Did these have any effects on the change in cognitive function and MRI findings?

4)      Were patients screened for AF recurrences after the PVI procedure?

5)      I have some trouble with the sentence on page 8, lines 311-313 (…significant restoration of cardiac output was not clear…). According to the baseline characteristics, the mean cardiac output was within normal range. Therefore, an improvement after PVI was rather unlikely. As a result, this study can’t exclude, that an increase in cardiac output could be another important mechanism in AF patients that suffer from concomitant heart failure with impaired cardiac output. This aspect could be addressed in the limitations section.  

Author Response

This interesting manuscript by Tatewaki et. al investigated the impact of pulmonary vein isolation (PVI) on cognitive-psychological function as well as regional cerebral blood flow (rCBF) and brain microstructures. The authors find an increased rCBF in the left posterior cingulate gyrus, and decreased cortical thickness in the left posterior insular cortex, accompanied by restored cognitive function, 6 months after PVI procedure. Thus, the authors conclude, that autonomic system alterations after catheter ablation may be a potential mechanism explaining the positive effects on cognitive function.

Despite its small size, this is an interesting study, which generates a valid hypothesis.

I have some comments, that should be addressed by the authors in a revised version of the manucript:

Author’s Response:

We appreciate the reviewer’s valuable important comments. We have revised our manuscript in accordance with the reviewer’s comments.

1)      Could the authors provide information about the exact duration of the AF disease? Did all patients profit equally from the procedure, regardless of the AF duration?

Author’s Response:

Thank you for the reviewer’s comment. We added the information of duration of AF in the result section.

Of the eight patients with AF, regarding the time from diagnosis to AF-ablation, 6 were less than one-year and 2 were one to three years in duration.”

      Regarding the differences in efficacy of AF ablation on cognitive function and regional CBF, only 2 patients had comparatively long AF duration of one to three years, but the corresponding data showed that the degree of improvement in cognitive function and in regional CBF in the left posterior cingulate gyrus were comparable to those of patients with AF duration of less than one year. In the future, we would like to obtain increase the number of AF patients’ data to conduct a detailed stratified analysis by AF duration.

2)      Could the authors povide information about the medication of the patients, particularly possible anti-arrhythmic drugs, anticoagulation and antidepressives? Did any patients receive additional anti-arrhythmic drugs after the PVI procedure.

Author’s Response:

Thank you for the reviewer’s comments. We inserted the following sentences in the result section.

 “At baseline, 5 patients were prescribed anticoagulants including warfarin and direct oral anticoagulant, and 1 was on antiplatelet drug, and 2 were on anti-arrhythmic medications. No patients were prescribed antidepressants.

We interviewed 7 patients who were available for post-intervention long-term follow-up. As the reviewer pointed out, antiarrhythmic drugs were prescribed for three of them, including two patients who relapsed more than six months after AF-ablation. The 5 patients were continuously on DOAC after ablation therapy.

3)      Did any patients have AF recurrences after the PVI treatment? Did these have any effects on the change in cognitive function and MRI findings?

Author’s Response:

    Thank you for the reviewer’s thought-provoking comments. The patients are followed regularly at the cardiology clinic against AF recurrence or rate control for recurrent AF. Unfortunately, after 6 months of treatment, they did not undergo follow-up cognitive tests or MRI scans. The long-term effects of ablation therapy on cognitive function are of interest and need to be explored.

4)      Were patients screened for AF recurrences after the PVI procedure?

Author’s Response:

    Thank you for the reviewer’s comments. As mentioned above, the patients were continuously followed-up at cardiology clinics. The two patients relapsed AF more than six months after AF-ablation therapy and have not recovered to sinus rhythm yet.

5)      I have some trouble with the sentence on page 8, lines 311-313 (…significant restoration of cardiac output was not clear…). According to the baseline characteristics, the mean cardiac output was within normal range. Therefore, an improvement after PVI was rather unlikely. As a result, this study can’t exclude, that an increase in cardiac output could be another important mechanism in AF patients that suffer from concomitant heart failure with impaired cardiac output. This aspect could be addressed in the limitations section.  

Author’s Response:

    Thank you for the reviewer’s important comments. We agree with the reviewer’s suggestion. This study can’t exclude the possibility that an increase in cardiac output could be an important mechanism in AF patients that suffer from concomitant heart failure with impaired cardiac output. We modified the description in the relevant part in the Discussion section as followed.

 “Although in our patients, significant restoration of cardiac output was not clear because the baseline of their cardiac output was not impaired, some cognitive domains and rCBF in the left retrosplenial cortex were improved after AF ablation, independent of cardiac output. Therefore, we speculate that AF-related adverse factors as well as cardiac output may be involved in the recovery of cognitive function after ablation therapy. Therefore, we speculate that AF-related adverse factors as well as cardiac output may be involved in the recovery of cognitive function after ablation therapy.

In the Limitations section, we inserted the following sentences,

The patients enrolled in this study have normal cardiac output at baseline. Therefore, this study could not examine the possibility that an increase in cardiac output could be an important mechanism of Ablation-depending cognitive recovery in AF patients that suffer from concomitant heart failure with impaired cardiac output.”
